# Challenges in Biodiversity Conservation in a Highly Modified Tropical River Basin in Sri Lanka

**Thilina Surasinghe** [1,*], **Ravindra Kariyawasam** [2], **Hiranya Sudasinghe** [3] and
**Suranjan Karunarathna** [4]

1   Department of Biological Sciences, Bridgewater State University, Dana Mohler-Faria Science & Mathematics Center, 24 Park Avenue, Bridgewater, MA 02325, USA
2   Center for Environment & Nature Studies, No.1149, Old Kotte Road, Rajagiriya 10100, Sri Lanka; waterforlifesl@gmail.com
3   Evolutionary Ecology & Systematics Lab, Department of Molecular Biology & Biotechnology, University of Peradeniya, Kandy 20400, Sri Lanka; hsudasinghe@gmail.com
4   Nature Explorations & Education Team, No. B-1/G-6, De Soysapura Flats, Moratuwa 10400, Sri Lanka; suranjan.karu@gmail.com
*   Correspondence: tsurasinghe@bridgew.edu; Tel.: +1-508-531-1908

**Abstract:** Kelani River is the fourth longest river in the South-Asian island, Sri Lanka. It originates from the central hills and flows through a diverse array of landscapes, including some of the most urbanized regions and intensive land uses. Kelani River suffers a multitude of environmental issues: illegal water diversions and extractions, impoundment for hydroelectricity generation, and pollution, mostly from agrochemicals, urban runoff, industrial discharges, and domestic waste. Moreover, loss of riparian forest cover, sand-mining, and unplanned development in floodplains have accentuated the environmental damage. In this study, based on Kelani River basin, we reviewed the status of biodiversity, threats encountered, conservation challenges, and provided guidance for science-based conservation planning. Kelani River basin is high in biodiversity and endemism, which includes 60 freshwater fish species of which 30 are endemic. Urbanization related threats are more severe in the middle and lower reaches while agriculture and impoundments peril the river in upper reaches. Documentation of these threats can be dated back to 1980, yet, Sri Lankan government has failed to take substantial actions for sustainable management of Kelani River basin, despite the presence of nearly 50 legislations pertaining to water and land management. Given high biodiversity richness, human dependency, and evident ecological deterioration, Kelani River basin should be prioritized for biodiversity conservation and sustainable resource management. Conservation and wise use of freshwater resources is a global concern, particularly for developing nations in Asia. Therefore, our review and guidance for scientifically informed conservation would serve as a prototype for basin-wide river management for Sri Lanka as well as for other developing nations of tropical Asia.

**Keywords:** aquatic habitat; freshwater fish; management; public awareness; water pollution; threats

## 1. Introduction

Freshwater is an indispensable, limited resource for the persistence of life. Freshwater ecosystems, particularly rivers, have acted as magnets for human settlements since ancient civilizations; consequently only a handful of river systems remains unaffected by human activities [1]. Anthropogenic stressors on rivers are disproportionately high in southern Asia, where more than half of the global megacities are located [2–5]. Increasing population growth and poverty, water pollution, and unsustainable water use have compounded the anthropogenic pressure on South Asian riverscapes [6–9].

In addition, river impoundments, channelization and diversions, and riparian deforestation have exacerbated this crisis [4,5,8,10]. Adversities of human impacts on river ecosystems have been substantially explored in Continental South Asia [2,4,8,10–12]. These studies have identified causes and effects of river pollution and provided recommendations for best management practices [13]. These studies are of utmost importance since access to clean water and sound management of aquatic resources are critical for higher living standards, improved human health, and greater economic growth [11,14]. As a developing nation rich in riverine biodiversity, the conservation and management of riverscapes is an urgent need on the island of Sri Lanka (65,610 km$^2$). Yet, the country lacks an ecologically informed plan for conservation and management of freshwater resources.

There are 103 major rivers in Sri Lanka; the island is part of the Western Ghats-Sri Lanka Biodiversity Hotspot [15] and contains a diverse and endemic biota including ≈950 vascular plants, 100 amphibians, 50 freshwater fish, and 50 freshwater crabs [16–18]. In the absence of natural lakes, the island's aquatic biodiversity (≈50 aquatic birds, ≈20 anurans, ≈15 reptiles, 90 freshwater fish) depends on its rivers, 16 of which are >100 km in length, accounting for 80% of the island's freshwater discharge [19–24]. Industrial and agricultural development, as well as urbanization, have led to substantial land-use transformations across Sri Lanka's major river basins, damaging hydrological variability, habitat structure, and water quality [25–27]. A sizable proportion of studies on conservation and management of Sri Lankan freshwater resources focuses on lentic systems [28–33], principally the ≈3000 reservoirs, 2 ha to 80 km$^2$ in extent, some of them two millennia old, which support agriculture in the arid and seasonal northern and eastern regions of the island. Besides, much information (>90%) on overall biodiversity of Sri Lanka's riverscapes, the threats they face, and the conservation challenges they present, is found in technical reports and conference proceedings—a poorly supported grey literature, which is less accessible to scientists, resource managers, and decision makers [27].

Among major rivers of Sri Lanka, the Kelani is remarkable. Ranked the fourth longest river on the island (145 km), it drains an area of 2314 km$^2$ and ranges from mean sea level to over 2300 m in altitude (Figure 1) [34,35]. The river meanders through multiple land-use land-cover types: forested central highlands in low-order upper reaches, low-moderate intensity development in its mid-reaches, and Sri Lanka's capital (Colombo) on its lower reaches (Figure 2) [35,36]. Kelani river is the principal consumable water source for 80% (over 6 million) of the human population of Colombo district [37]. Located in Sri Lanka's wet zone, the basin receives an annual average rainfall of 3400 mm corresponding to 7800 million cubic meters in total annual volume [38]. Here we review the indigenous freshwater-fish diversity and the threats this fauna faces, and provide recommendations for the conservation and management of the Kelani river ecosystem based on the published literature and our own observations.

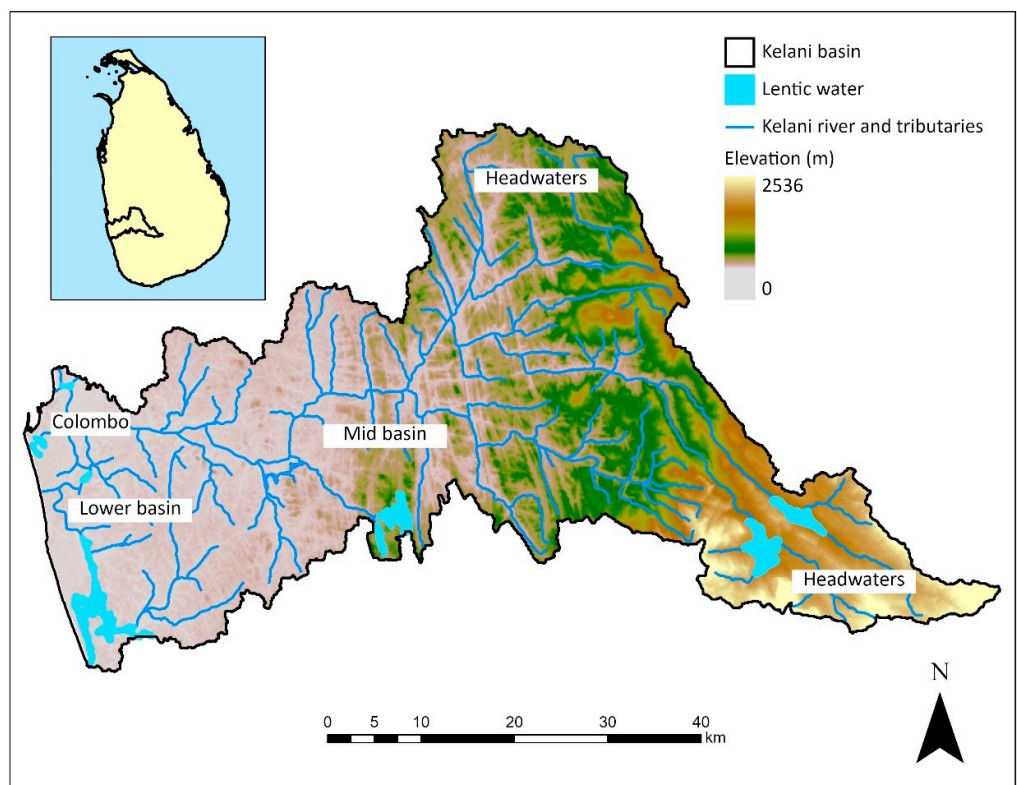

**Figure 1.** Kelani River Basin of Sri Lanka (only the mainstem, major tributaries, and associated lentic systems are shown). The commercial capital of Sri Lanka (Colombo), the upper, lower, and mid-basins are also indicated.

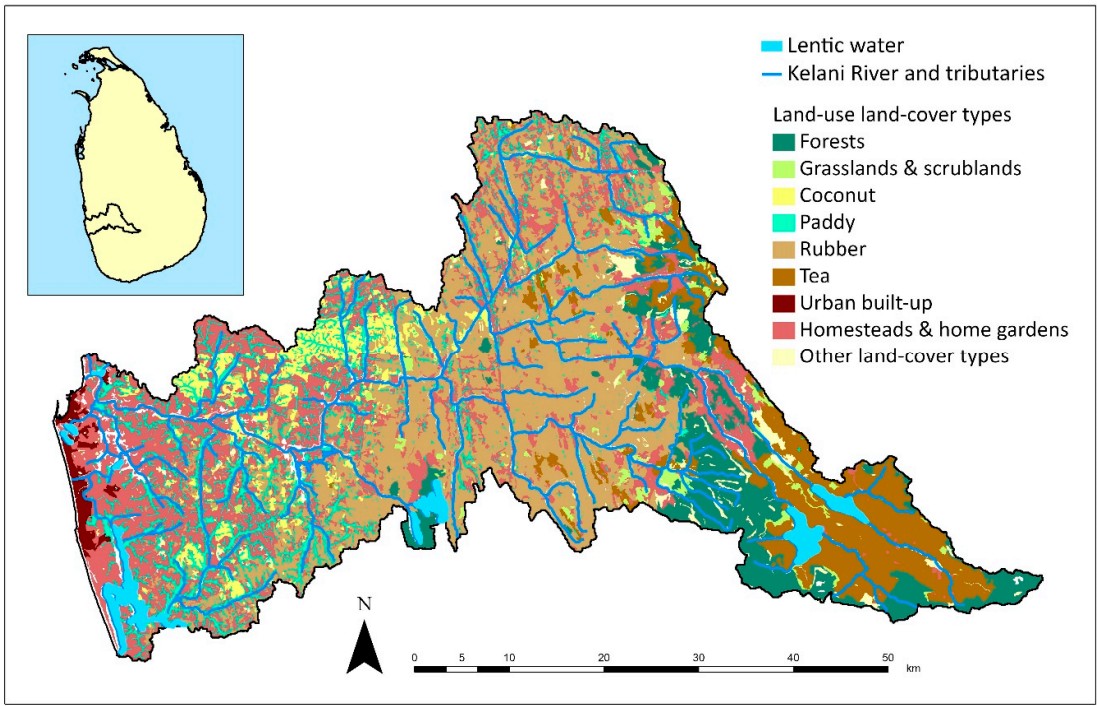

**Figure 2.** Land-use and land-cover types of the Kelani basin. Data sources: Sri Lanka Survey Department 2015.

## 2. Threats for the Kelani River

### 2.1. Loss of Riparian Vegetation and Land-Cover Modifications in the River Basin

Urbanization, industrial development, and agriculture are largely responsible for land-cover transformations in the Kelani River basin (Figure 2) [7,39]. Only 10% of the basin remains forested, much of the forest loss is likely have occurred after the colonial period [40,41]. Urbanization and infrastructure development were the primary drivers of deforestation in the lower basin, while agriculture was responsible for declining forest cover in the upper and mid-basins [9]. The forest cover (24,090 ha) of the Kelani basin is compressed to the upper basin, consisting of montane and sub-montane moist evergreen forests [40]. Although some riparian vegetation exists in Colombo's outskirts, the vegetation structure is limited to short grasses, shrubs, and scrub; large mature trees with dense canopy are mostly absent (RK, personal observations). Rivers are intricately linked with their watersheds, particularly the riparian zone. The trophic dynamics of streams are driven by the exogenous supply of terrestrial leaf litter as well as upland arthropods [42,43] while streamside vegetation moderates stream temperature, filters sediments and nutrients from surface runoff, and supplies large woody debris to increase channel complexity [1]. Loss of riparian forests inevitably leads to changes in a river's thermal properties and nutrient loading, and the homogenization of channel structure.

### 2.1.1. Urbanization

The Kelani River flows through metropolitan Colombo—the largest city in the country (population 6.6 million, density 134,680 $km^{-2}$), with the highest coverage of impervious surfaces ($\approx$40% urban built-up land cover), and burgeoning industries and businesses—where the river is vulnerable to synergistic threats [40,44,45]. As Colombo's suburbs expand, the basin is undergoing major land-cover transformations [46]. Over two decades (1989–2016), the built-up environment of Colombo metropolis increased by 25% while forest cover and freshwater habitats (including wetlands) declined by 6% and 52%, respectively [41,47]. Homesteads cover 25% of the Kelani basin; shanties and slums occupy much of the floodplain closer to the river mouth (Figure 3) [9,40].

### 2.1.2. Agriculture

In the middle and upper basin, commercial-scale agriculture has encroached on the floodplains and riparian zone. A third of the Kelani basin is covered by rubber plantations while 10% is occupied by tea plantations; a substantial extent (7.8% of the basin) of rice paddies are found in the mid and lower basins, whereas coconut plantations account for a lower proportion (5%) [9,40,48]. Plantation agriculture, particularly tea, is responsible for extensive forest loss in the upper watershed of Kelani River (Figure 3) [20,49].

### 2.1.3. Wetland Loss

Historically, much of the Kelani floodplain was speckled with a variety of wetlands [35]. These have since been drained, filled, or dredged for urban development or for agriculture (rice fields), greatly diminishing the extent of floodplain wetlands and freshwater marshes; consequently <1% of the basin is now occupied by wetlands [20,35]. These wetlands served multiple ecological functions, including flood mitigation and flood storage, nutrient assimilation and denitrification, silt retention, groundwater recharge, and provision of wildlife habitats, drought refuges, and breeding and nursery grounds [49–52].

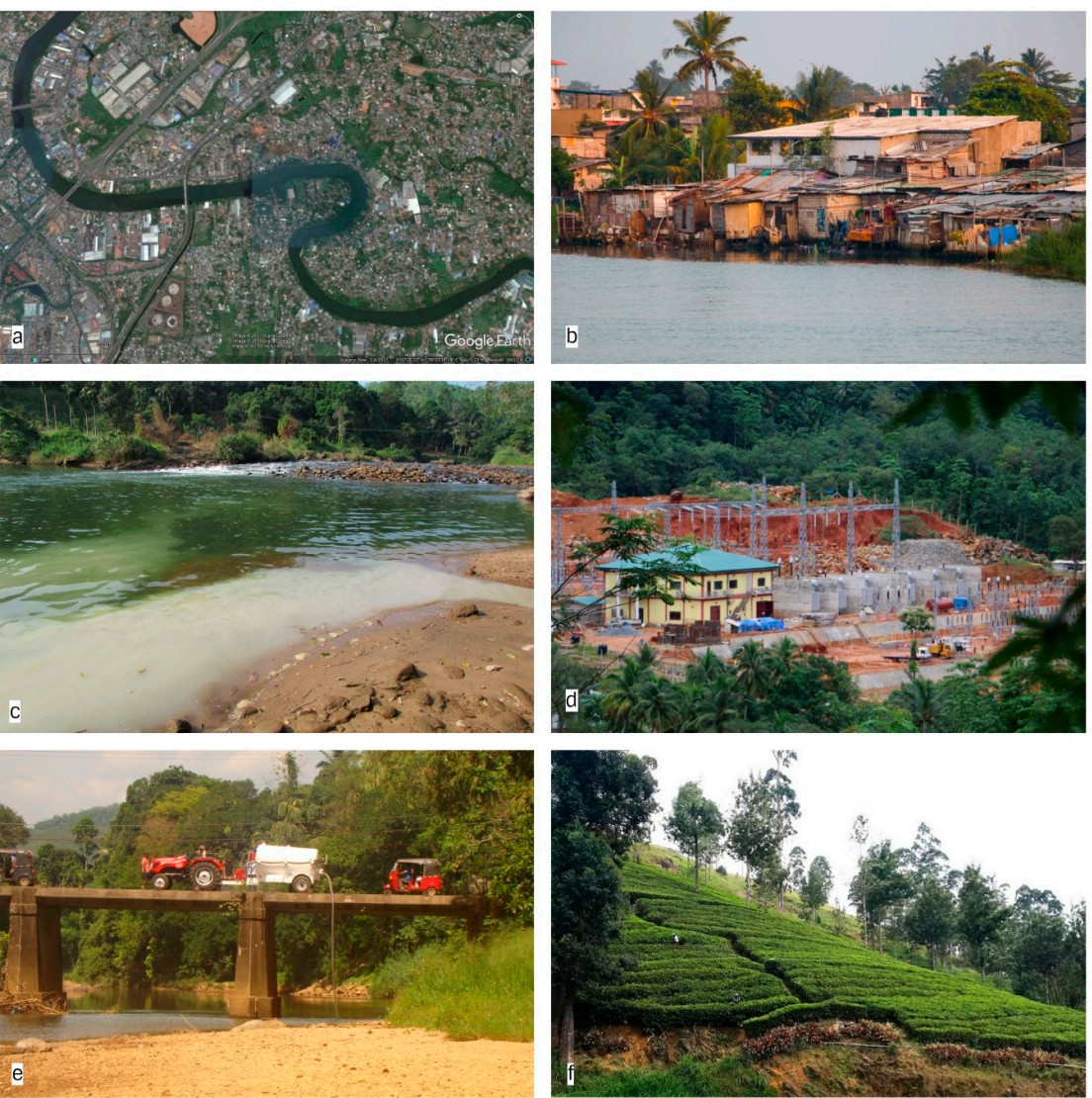

**Figure 3.** Threats observed in the Kelani River basin: (**a**) urban development in the riparian zone in the lower reaches (GoogleMaps, 2017); (**b**) shanties and slums constructed on the edge of the river channel (Dhammika Heenpella); (**c**) pollution with sewage effluvia; (**d**) a hydropower plant under construction; (**e**) illicit extraction of water from the river; (**f**) commercial-scale tea plantations in close proximity to the river channel (Hiranya Sudasinghe, Suranjan Karunarathna, Senanayaka Bandara).

## 2.2. Water Pollution

Water pollution in the Kelani River stems from numerous sources—both non-point sources as well as point sources [27]. The extent of pollution strongly correlates with population density across the watershed. In the lower basin, where 25% of the national population resides (>6 million, density > 2500 $km^{-2}$), the integrated pollution index consistently underperformed water-quality standards for consumption, recreation, and aquatic biodiversity [37]. In the coastal reaches, biological oxygen demands (BOD) have reached 17 mg $L^{-1}$, while pH has fallen to 5.3, with higher conductivity (>0.2 ms $cm^{-1}$); total colifom counts as high as 1500 cells 100 $mL^{-1}$ have been reported even in the middle reaches [37]. Although nutrient spiraling and flushing remediate pollution during the high-discharge wet season (800–1800$m^3$ $s^{-1}$), pollutant resident time can be much higher during low-flow dry (20–25$m^3$ $s^{-1}$) season [38,53].

### 2.2.1. Point-Source Pollution

Throughout lower and mid-reaches of the river, unregulated waste disposal from industrial and thermal effluvia, and stormwater, have been serious issues, particularly in Colombo metropolis since the 1980s (Table 1, Figure 3) [7,9,31,36]. At least four major municipalities within the Kelani basin discharge untreated or partially-treated sewage and domestic wastewater into the river [27,54–57], leading to excessive accumulation of organic waste, which then increases the BOD [58]. Over 6000 industries directly discharge waste into river (Table 1) [9,31,39,59]. The average daily point-source discharge is estimated to be over 414,600 cm$^3$, creating a biological oxygen demand in excess of 11,600 kg day$^{-1}$ [59]. Although the focal environmental regulator of Sri Lanka, the Central Environmental Authority, identified these pollution sources decades ago, neither legal nor remedial actions have been taken thus far [31]. The contents of these industrial discharges include thermal effluvia, oils, petroleum impurities, heavy metals, and synthetic organic compounds such as polychlorinated biphenyls [27,60–62]. Some of these contaminants are carcinogens (4-methylimidazole) or endocrine disrupters (Butyltin) [60,63]. Chlorinated organic compounds have bioaccumulated in the tissues of fish and invertebrate inhabiting the lower reaches of the Kelani; these can induce chronic immunodeficiency and histological malformations [63].

**Table 1.** Sources and types of pollution in Kelani River based on field observations and literature survey.

| Pollutant | Source of the Pollutant |
| --- | --- |
| Sewage and other organic compounds | Municipalities, slums, hotels, hospitals, restaurants, water and sewage treatment plants, breweries, leather tanneries, plywood factories, rubber and latex factories, beverage and food processing plants |
| Heavy metals and other inorganic compounds | Tanneries, petroleum refineries, metal processing and manufacturing plants, battery manufacturing |
| Detergents | Hotels, households, breweries |
| Fabric dye | Export processing zones, textile and garment factories |
| Synthetic organic compounds | Brewery, wood processing plants, beverage factories, rubber and plastic factories, wood processing industries |
| Petroleum wastes and oil | Petroleum and oil refineries, transportation businesses, motor vehicle service stations |
| Synthetic inorganic compounds | Fertilizer manufactures, beverage factories, lather tanning, rubber and plastic factories, tire manufactures |
| Medical and pharmaceutical waste | Open waste disposal sites, hospitals and medical centers |
| Solid waste | Households, businesses and commercial enterprises, construction sites, garbage disposal sites |

### 2.2.2. Non-Point Source Pollution

Non-point source pollution in Kelani River during the monsoon season is noteworthy, particularly from stormwater canals that have drained into the river since the 1990s [54]. By the year 2000, the growing number of open solid-waste disposal sites have compounded non-point source pollution in Kelani River [27]. The urban runoff contains many biologically hazardous contaminants including heavy and trace metals, oils, and hydrocarbons, an issue well documented in both current (2007) [59] and historical (1985) [61] contexts. Consequently, elevated levels of conductivity (0.006–0.009 Sm$^{-1}$), chemical (11.8–19.4 mg L$^{-1}$) and biological (1.7–2.9 mg L$^{-1}$) oxygen demand, and total coliform bacteria (30,600–51,000 cells 100 mL$^{-1}$) were recorded at both industrial and residential reaches of the Kelani River, while certain urban reaches failed to meet drinking-water quality standards [57,64].

Expansion of agriculture in the river basin has also resulted in water quality degradation. For instance, watershed-wide deforestation exacerbates soil erosion and subsequent sedimentation of the river channel, leading to flashfloods in the wet season [9]. Estimated soil erosion for Kelani Basin varied widely (0–103.69 t ha$^{-1}$ yr$^{-1}$) with a mean of 10.88 t ha$^{-1}$ yr$^{-1}$, according to a Revised Universal

Soil Loss Model [65]. The mid basin (25% of the entire basin) is likely to suffer the most severe erosion (12–25 t ha$^{-1}$ yr$^{-1}$), exceeding the maximum tolerable soil loss (9–13 t ha$^{-1}$ yr$^{-1}$). Consequently, mid reaches are likely to experience the greatest eroded sedimentation flux. Moreover, nearly 8280 mt of nitrogen and phosphate fertilizers are applied annually to tea and rubber plantations and rice fields of the basin [35,60]. Ammonia concentration in coastal reaches averages around 0.22 mg L$^{-1}$, which is closer to maximum permissible (0.2 mg L$^{-1}$) levels [60]. Elevated phosphate and nitrate concentrations in the river from agricultural runoff have led to eutrophication and algal blooms in sluggish parts of the river, particularly during low-flow seasons [27,35,61].

### 2.3. Hydrological Changes and Overexploitation

The Kelani River's hydrology is modified by reservoir impoundments. By 2016, five major hydropower reservoirs (constructed since 1950) and 32 mini-hydropower plants (since 2000) impeded the river's flow, in addition to several flood levees and dikes [34,66,67]. Another hydropower plant is under construction at the upper reaches of Kelani, while proposals are being drafted for several more [66] (Figure 3). Such flow obstructions result in modified discharge regimes, flood pulses, and water chemistry (particularly conductivity, dissolved oxygen, and alkalinity), which in turn changes the habitat structure of both the river channel and the floodplains [5,68]. In certain Kelani tributaries, 60% of the tributary length have become either dead or low-flow reaches below mini-hydropower dams; nearly 30% of the course of Kelani River is estimated to be impaired due to impoundments [66,67]. For instance, dams cut off sediment and allochthonous organic matter supply to below-dam reaches [62]. Impoundments shift high-discharge cold-water perennial lotic systems into deep, warm-water lentic systems upstream of the dam, while the tailwater becomes a low-discharge seasonal lotic system; neither post-impoundment habitat is suitable for fluvial-adapted biota [10,62]. Moreover, reservoirs are less suitable as spawning grounds for fish and macroinvertebrates [10]. Marked changes in the fish community structure have been documented in the Kelani River where the dominant species in tailwater changed from cyprinids to cichlids [62]. Hydrologic modifications can also imperil populations of endemic, range-restricted, and threatened montane fish [66]. Two recently proposed hydropower projects (Broadland hydropower plant and Seethawaka River mini-hydropower plant) in the Kelani basin are likely to threaten the remnant populations of *Systomus asoka* and other endemic fish [66].

The growing population (Figure 4) and rising living standards have led to an increase in per capita water demand in Colombo metropolis and other municipalities within the Kelani basin. During drought years, increasing demand dramatically attenuate discharge. Numerous illicit water diversions and extractions for irrigation and consumption occur throughout the basin [7,42]. Moreover, illicit sand and gem mining (dredging) during low-flow periods drastically change the channel geomorphology, streambed microtopography, and water chemistry [20,42,49]. Primarily attributed to unsustainable water withdrawal and sand mining, seawater intrudes 15 km (in contrast to 0.7 m mean spring tide range in the region) inland along the Kelani River since early 1990s, particularly during low-discharge (<33 m$^3$ s$^{-1}$) droughts [35]. In addition, removal of small boulders from the river channel for landscaping and woody-debris removal to make way for navigation and recreational activities adversely impact habitat structure and aquatic food webs, as those substrates provide both habitats and invertebrate-rich feeding grounds for fish [42,62].

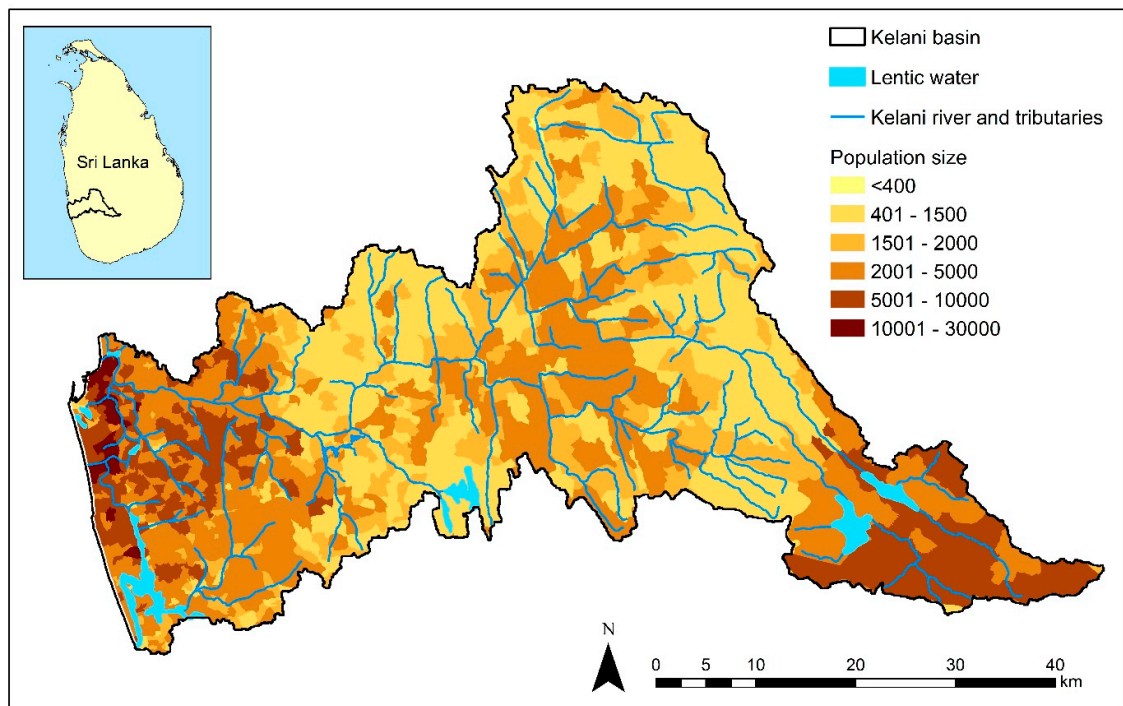

**Figure 4.** Human population size of census blocks within Kelani River basin. Data Source: Department of Census and Statistics Map Portal, Sri Lanka [48].

*2.4. Policy Issues*

Although there are some 20 governmental agencies and over 50 legal instruments regulating aquatic resources in Sri Lanka, lack of proper coordination among these agencies have weakened the effective enforcement (Table 2) [54,57,69–73]. Conflicting interests among some regulatory statutes—for instance, Land Reclamation and Development Cooperation Act versus National Policy on Wetlands—further contribute to policy dissociations. Similarly, the Water Resource Board Act and the National Water Supply and Drainage Board Act only address issues germane to drinking water; they do not contain provisions for land management across the watershed [74]. None of these policies facilitate or encourage inter-agency collaboration or partnerships between local administrative units, hampering both conflict resolution and effective collaboration between stakeholders.

Further, implementation of the policy stipulations is also unsatisfactory. For instance, although National Environmental Act requires an environmental impact assessment for major development activity that impacts the environment, no evidence exists for mitigatory or remedial actions taken against pollution in Kelani River. No mechanisms exist to ensure post-developmental monitoring and developer's adherence to mitigatory and preventive strategies [55].

Despite recent efforts to identify stakeholders of the Kelani River basin, no steps have been taken for effective stakeholder networking for multidirectional knowledge sharing, enforcement, and capacity building [9,12]. Even recent legislation is not explicit with regard to integrated river basin management and do not effectively harness novel conservation designs. While mandatory streamside reservations (20–60 m) [36] are often overshadowed by short-term economic goals, the ecological reasoning for these buffer widths remain unjustified [71]. Furthermore, although Sri Lanka is a signatory to many international conventions (Ramsar Convention and UN Waters), current policies have not drawn insights from such international collaborations.

**Table 2.** Environmental regulations of Sri Lanka that relate to conservation and management of the Kelani River basin.

| Statute Name | Issues Addressed Related to Inland Aquatic Resources | Enforcing Agency | Enactment Year (Last Amendment) |
|---|---|---|---|
| **Overarching Environmental Statues** | | | |
| National environmental Act | Provisions on environmental management, protection, and monitoring. Approval of projects via overseeing environmental impact assessment methods. | Central Environmental Authority | 1980 (2000) |
| National Environmental Policy and Strategies | Provides directions to conserve and manage environment in all aspects, including forestry and wildlife conservation, fisheries and other aquatic resources, particularly with respect to agricultural sustainability. | Ministry of Mahaweli Development and Environment | 2003 |
| Cleaner Product Policy | Improve efficiency of water and energy consumption by minimizing wastage and over exploitation. | Ministry of Mahaweli Development and Environment | 2005 |
| National Forest Policy Forest Conservation Ordinance Amendment Act | Conserve forests for their biodiversity, soils, water and historical, cultural, religious, and aesthetic values. Increase the tree cover and forest productivity. Enhance contribution of sustainable forestry to rural development and national economy. | Forest Conservation Department | 1995 1907 (2009) |
| National Wildlife Policy Fauna and Flora Protection Ordinance Amendment Act | Conserve aquatic biodiversity and habitats to prevent misuses. Maintain ecological processes and genetic diversity. Delineate conservation lands, inclusive of embed aquatic systems. Engage local communities in protected-area management. | Department of Wildlife Conservation | 2000 1937 (2009) |
| **Watershed or Basin-Scale Statues** | | | |
| National Watershed Management Policy | Conserve, restore, and manage watersheds while managing their critical environmental dynamics. | Ministry of Mahaweli Development and Environment | 2004 |
| National Land-use Policy National Policy on Protection and Conservation of Water Sources, their Catchments and Reservations in Sri Lanka | Design land-use policies at watershed scale. Conserve and restore all water sources, including watersheds as a reservation. Promote sustainable water consumption through participatory management of responsible governmental institutions and communities. Identify needs for regulation reforms. Remedy shortcoming of existing policies and promote watershed-wide management of inland aquatic resources. | Ministry of Land and Land Development | 2007 2014 |

**Table 2.** *Cont.*

| Statute Name | Issues Addressed Related to Inland Aquatic Resources | Enforcing Agency | Enactment Year (Last Amendment) |
|---|---|---|---|
| **Inland Aquatic Resources Conservation and Management Statues** | | | |
| National Policy on Wetlands | Conserve wetland ecosystems, to prevent illegal utilization. Restore and maintain the biological diversity and productivity. Enhance wetland ecosystem services. Assure sustainable use of wetlands by local communities. Achieve national commitments to the Ramsar Convention. | Ministry of Mahaweli Development and Environment | 2005 |
| Irrigation ordinance | Construct, maintain, and protect irrigation works, water conservation | Department of Irrigation | 1856 |
| National Drinking Water Policy | Provide a framework for sustainable and efficient supply of safe water in adequate quantity at an affordable cost. | Ministry of Water Supply and Drainage | 2002 |
| National Water Supply and Drainage Board Act | Provide safe drinking water and facilitate sanitation. | National Water Supply and Drainage Board | 1974 (1992) |
| Water Resource Board Act | Conserve and sustainably utilize aquatic resources by harnessing new technologies and management tools to meet the growing demands. | Water Resource Board | 1984 (1999) |
| Fisheries and Aquatic Resources Act | Manage, regulate, conserve, and develop the fisheries and aquatic resources, including any native aquatic organisms in any stage of their life cycle and non-living substances found in aquatic media | Department of Fisheries and Aquatic Resources | 1996 (2016) |
| **Land and Infrastructure Development Statues** | | | |
| Land Reclamation and Development Cooperation Act | Mitigate flood damage by restoring and maintaining pollution-free inland water bodies. Design basin-scale land development plans. Establish wetland reclamation standards and provisions against wetland filling. | Land | Land Reclamation and Development Cooperation Act |
| Urban Development Act | Promote integrated planning and implementation for environmental, economic, social, and physical development of urban areas | Urban Development Authority | 1978 |
| Flood Protection Ordinance | Protection, land-use regulation, and management of flood-prone areas, such as floodplains | Department of Irrigation | 1924 (1955) |
| Soil Conservation Act | Introduce provisions to conserve soil resources, prevent or mitigate soil erosion, and protect land against flood and drought related damage. Identify and regulate erodible areas. | Soil Conservation Board | 1951 (1996) |
| Land Development Ordinance Crown Lands Ordinance State Lands (recovery of possession) Act | Establish provisions for riparian forest encroachment by stipulating minimum buffer-width reservations. Regulation of aquatic resources in lakes and streams. | Ministry of Lands and Land Development | 1935 1947 1979 |

Nonetheless, the scientific reasoning behind Sri Lanka's existing policies and law execution are weak or nonexistent. A dearth of biologists has afflicted the Sri Lankan environmental agencies for decades, and consequently, existing Sri Lankan environmental policies failed to meet scientific rigor [57,71,72]. Therefore, biologists specialized across multiple subdisciplines (wildlife biologists, conservation biologists, freshwater and wetland ecologists, landscape ecologists, ecosystem ecologists, environmental pathologists, and epidemiologists) should be involved at the decision-making level.

## 3. Freshwater Biodiversity in Kelani River Basin

Kelani River is an ecosystem complex comprising a wide spectrum of habitats, ranging from mountain springs and ephemeral headwater streams to large perennial rivers as well as a diverse array of inland and coastal wetlands (Figure 5). Meandering through three major floristic regions—northern wet lowlands, foothills of Adam's Peak and Ambagamuwa, and the Adam Peak highlands—the Kelani River associates three diversity rich forest communities: tropical wet evergreens, tropical moist upper montane forests, and tropical submontane evergreens [75]. A critically endangered, narrow-range endemic woody species, *Balanocarpus kitulgalensis* is restricted to riparian forests of the Kelani basin [9]. The Kelani is also known for hosting narrow-range endemic invertebrates, including three species of critically endangered freshwater crabs. In addition, 16 and 23 nationally threatened dragonfly and butterfly species respectively, occur in the basin [21]. A total of 73 nationally threatened non-fish vertebrates (nine amphibians, 11 reptiles, 25 birds, and 28 mammals) are also associated with the basin [76].

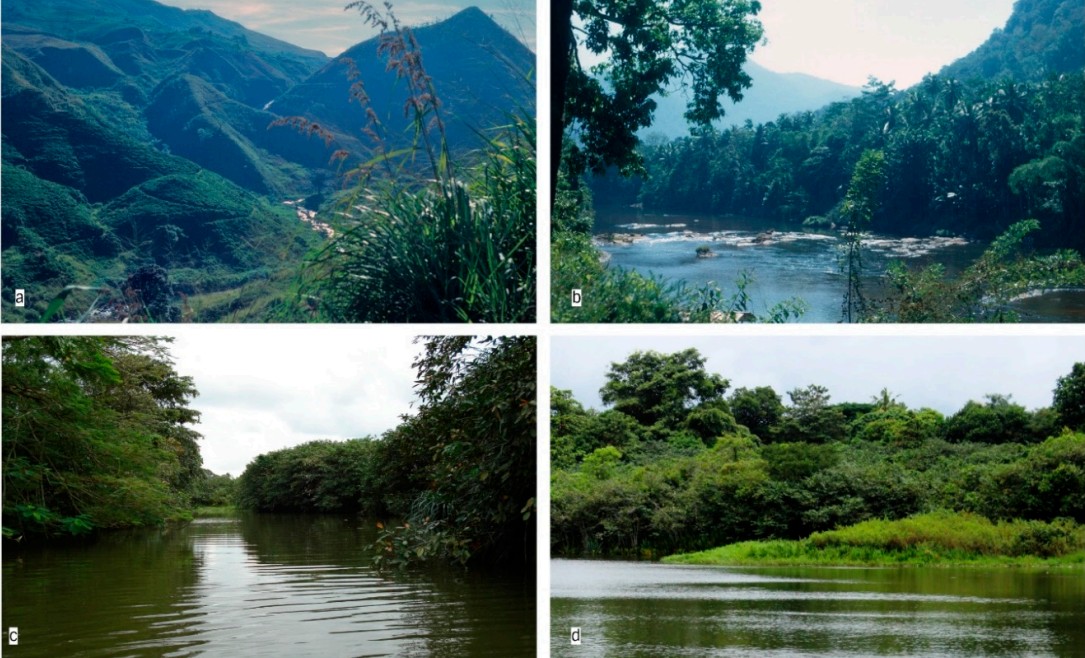

**Figure 5.** Habitats in different reaches of the Kelani River: (**a**) headwaters of central highlands (Duncan Campbell and Douglas Campbell); (**b**) mid-reaches, with forested riparian zone (Joanne Goldby); (**c**) inland wetland habitats; (**d**) coastal wetland complex at Beddagana Wetland Park (Kassapa Jayasinghe, Madhava Botejue).

*Freshwater Fish Diversity in Kelani River Basin and Threats Encountered by Fish Communities*

Scientifically informed conservation planning should identify and provide protection to native aquatic biodiversity, particularly freshwater fish. Thus, reviewing the diversity and distribution of freshwater fish is critical. Kelani River basin is nested within the southwestern ichthyological province of Sri Lanka where more than half of Sri Lanka's endemic freshwater fish occur [42,76]. Here, combining

literature review and authors' personal observations, we documented a total of 60 freshwater fish species in the Kelani basin, of which 30 are endemic (Table 3, Figure 6). This fish community accounts for 63% Sri Lankan freshwater fish diversity and represents 17 families. Among the fish community of the Kelani, 22 species are threatened (six Critically Endangered; 13 Endangered; three Vulnerable) [17].

**Table 3.** The list of indigenous freshwater fish species recorded in the Kelani River basin, Sri Lanka.

| Family | Species | Common Name | National Conservation Status | Information Source |
|---|---|---|---|---|
| Cyprinidae | *Amblypharyngodon grandisquamis*(SLE) | Silver Carplet | LC | FOA/MUR |
| Cyprinidae | *Dawkinsia singhala*(SLE) | Sri Lankan Filamented Barb | LC | FOA/MUR |
| Cyprinidae | *Devario malabaricus*(NTV) | Giant Danio | LC | FOA/MUR |
| Cyprinidae | *Devario micronema*(SLE) | | - | FOA/MUR |
| Cyprinidae | *Esomus thermoicos*(NTV) | Sri Lankan Flying Barb | LC | FOA/MUR |
| Cyprinidae | *Garra ceylonensis*(SLE) | Sri Lankan Stone Sucker | VU | FOA/MUR |
| Cyprinidae | *Horadandia atukorali*(SLE) | Horadandia | VU | FOA |
| Cyprinidae | *Labeo heladiva*(SLE) | Common Labeo | LC | FOA |
| Cyprinidae | *Laubuka varuna*(SLE) | Varuna Laubuka | CR | FOA/MUR |
| Cyprinidae | *Pethia bandula*(SLE) | Bandula Barb | CR | FOA/MUR |
| Cyprinidae | *Pethia nigrofasciata*(SLE) | Black Ruby Barb | EN | FOA/MUR |
| Cyprinidae | *Pethia reval*(SLE) | Red-finned Barb | EN | FOA/MUR |
| Cyprinidae | *Puntius bimaculatus*(NTV) | Red Side Barb | LC | FOA/MUR |
| Cyprinidae | *Puntius dorsalis*(NTV) | Long Snouted Barb | LC | FOA/MUR |
| Cyprinidae | *Puntius kamalika*(SLE) | Kamalika's Barb | EN | FOA/MUR |
| Cyprinidae | *Puntius kelumi*(SLE) | Kelum's Barb | EN | FOA/MUR |
| Cyprinidae | *Puntius chola*(NTV) | Swamp Barb | LC | FOA/MUR |
| Cyprinidae | *Puntius titteya*(SLE) | Cherry Barb | EN | FOA/MUR |
| Cyprinidae | *Puntius vittatus*(NTV) | Silver Barb | LC | FOA/MUR |
| Cyprinidae | *Rasbora dandia*(NTV) | Broad-lined Striped Rasbora | LC | FOA/MUR |
| | *Rasbora microcephalus*(NTV) | Narrow-lined Striped Rasbora | LC | FOA/MUR |
| Cyprinidae | *Rasboroides vaterifloris*(SLE) | Fire Rasbora | - | FOA |
| Cyprinidae | *Systomus pleurotaenia*(SLE) | Black Lined Barb | EN | FOA/MUR |
| Cyprinidae | *Systomus asoka*(SLE) | Asoka Barb | CR | FOA/MUR |
| Cyprinidae | *Systomus sarana*(NTV) | Olive Barb | LC | FOA/MUR |
| Cyprinidae | *Tor khudree*(NTV) | Mahseer | NT | FOA/MUR |
| Cobitidae | *Lepidocephalichthys thermalis*(NTV) | Common Spiny Loach | LC | FOA/MUR |
| Nemacheilidae | *Paracanthocobitis urophthalma*(SLE) | Tiger Loach | EN | FOA/MUR |
| Nemacheilidae | *Schistura notostigma*(SLE) | Banded Mountain Loach | NT | FOA/MUR |
| Bagridae | *Mystus gulio*(NTV) | Long Whiskered Catfish | LC | FOA |
| Bagridae | *Mystus nanus*(SLE) | Sri Lankan Dwarfed Striped Catfish | LC | FOA/MUR [77] |
| Bagridae | *Mystus ankutta*(SLE) | Sri Lankan Dwarf Catfish | EN | FOA/MUR |
| Bagridae | *Mystus zeylanicus*(SLE) | Sri Lankan Yellow Catfish | LC | FOA/MUR |
| Siluridae | *Ompok argestes*(SLE) | Sri Lankan Mottled Butter Catfish | DD | FOA/MUR [78] |
| Siluridae | *Wallago attu*(NTV) | Silver Shark | EN | FOA |

<div align="center"><b>Table 3.</b> <i>Cont.</i></div>

| Family | Species | Common Name | National Conservation Status | Information Source |
|---|---|---|---|---|
| Clariidae | *Clarias brachysoma*(SLE) | Walking Catfish | NT | FOA/MUR |
| Heteropneustidae | *Heteropneustes fossilis*(NTV) | Stinging Catfish | LC | FOA |
| Aplocheilidae | *Aplocheilus dayi*(SLE) | Day's Killifish | EN | FOA/MUR |
| Aplocheilidae | *Aplocheilus parvus*(NTV) | Dwarf Killifish | LC | FOA |
| Cichlidae | *Etroplus suratensis*(NTV) | Green Chromide | LC | FOA |
| Cichlidae | *Pseudetroplus maculatus*(NTV) | Yellow Chromide | LC | FOA |
| Gobiidae | *Awaous melanocephalus*(NTV) | Scribbled Goby | LC | FOA |
| Gobiidae | *Glossogobius giuris*(NTV) | Bar Eyed Goby | LC | FOA |
| Gobiidae | *Schismatogobius deraniyagalai*(NTV) | Red Neck Goby | EN | FOA |
| Gobiidae | *Sicyopterus griseus*(NTV) | Stone Goby | CR | FOA/MUR |
| Gobiidae | *Sicyopus jonklaasi*(SLE) | Lipstick Goby | EN | FOA |
| Anguillidae | *Anguilla bicolor*(NTV) | Level Finned Eel | LC | FOA |
| Anguillidae | *Anguilla nebulosa*(NTV) | Long Finned Eel | LC | FOA |
| Anabantidae | *Anabas testudineus*(NTV) | Climbing Perch | LC | FOA |
| Synbranchidae | *Monopterus desilvai*(SLE) | Sri Lankan Lesser Swamp Eel | CR | FOA |
| Synbranchidae | *Ophisternon bengalense*(NTV) | Asian Swamp Eel | CR | MUR |
| Osphronemidae | *Belontia signata*(SLE) | Combtail | NT | FOA/MUR |
| Osphronemidae | *Pseudosphromenus cupanus*(NTV) | Spike Tailed Paradisfish | LC | FOA/MUR |
| Channidae | *Channa ara*(SLE) | Sri Lankan Giant Snakehead | EN | FOA |
| Channidae | *Channa gachua*(SLE) | Brown Snakehead | LC | FOA |
| Channidae | *Channa orientalis*(SLE) | Smooth Breasted Snakehead | VU | FOA |
| Channidae | *Channa punctata*(NTV) | Spotted Snakehead | LC | FOA |
| Channidae | *Channa striata*(NTV) | Murrel | LC | FOA |
| Mastacembelidae | *Mastacembelus armatus*(NTV) | Marbled Spiny Eel | LC | FOA/MUR |
| Belonidae | *Xenentodon cancila*(NTV) | Freshwater Garfish | NT | FOA |

Abbreviations: SLE—Endemic to Sri Lanka; NTV—Native; CR—Critically Endangered; EN—Endangered; VU—Vulnerable; LC—Least Concerned; NT—Near Threatened; DD—Data Deficient; FOA—Field observation by Authors; MUR—Museum records.

Among the freshwater fishes of Kelani River basin, two are microendemics: *Pethia bandula* (Bandula barb) and *Systomus asoka* (Asoka barb, Figure 5). The former is restricted to a narrow stretch of single tributary (at Galapitamada, Kegalle district) in the middle reaches of the basin; the latter have been documented in several tributaries of the Kelani river in the foothills (Kithulgala and Deraniyagala, Kegalle District) [32,79]. The two species of Synbranchid eels (swamp eels), *Monopterus desilvai* (Lesser swamp eel) and *Ophisternon bengalense* (Asian Swamp eel) are found in the coastal swamps hydrologically connected to the lower reaches of Kelani River. Filling such wetlands threatens these eels.

Fish species specialized for highly-oxygenated fast-flowing cold waters such as *Garra ceylonensis* (Ceylon stonesucker), *Systomus pleurotaenia* (Black lined barb), and *Schistura notostigma* (Banded mountain loach) cannot tolerate hydrological modifications such as channelization, impoundments, and pollution [62]. *Pethia nigrofasciata* (Black ruby barb) prefers canopy-shaded streams and are unlikely to persist where riparian forest is lost. Due to industrial and domestic waste discharge in lower reaches, occasional fish kills have been reported in the last few decades [57]. Streambed siltation stemming from watershed-scale deforestation may degrade and destroy nesting and spawning grounds. *Puntius titteya* (Cherry barb, Figure 5), an endemic fish found in shady, slow-flowing streams, still occurs even in the suburbs of Colombo [80]. However, its habitats are being degraded and is becoming increasingly uncommon.

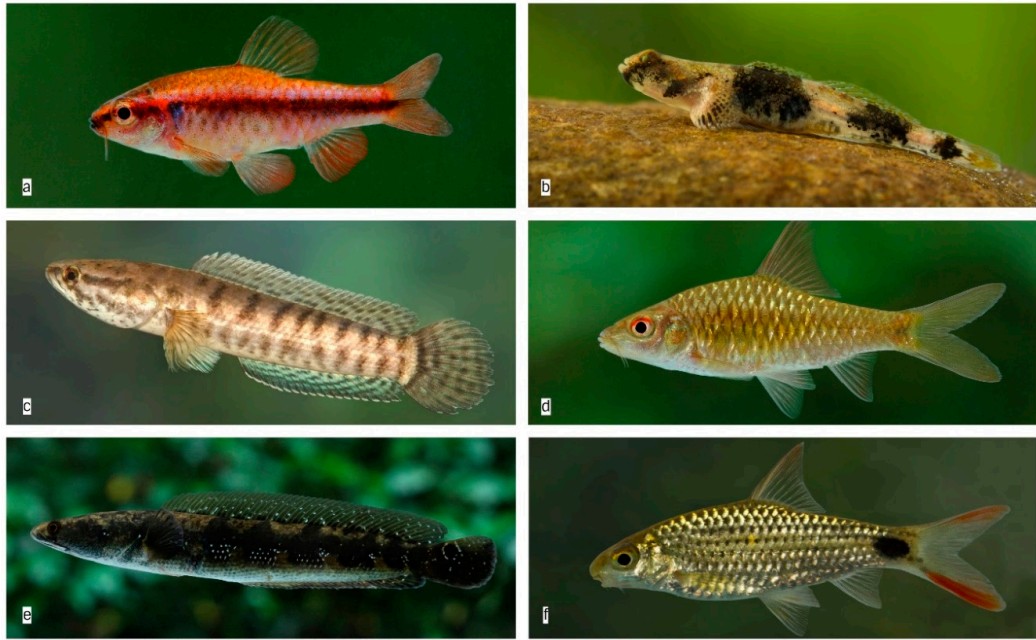

**Figure 6.** Native freshwater fish species found in Kelani River basin (**a**) *Puntius titteya*; (**b**) *Schismatogobius deraniyagalai*; (**c**) *Channa punctata*; (**d**) *Puntius kelumi*; (**e**) *Channa ara*; (**f**) *Systomus asoka* (Mahesh De Silva, Hiranya Sudasinghe, Sameera Akmeemana, Krishan Wewalwala).

Several cyprinid species of the Kelani River migrate upstream for breeding during the monsoon season; impoundments may interrupt their life cycles and negatively impact recruitment [10]. Diadromous fish of Kelani River, such as sicydiine gobies *Sicyopus jonklaasi* (Lipstick goby) and *Sicyopterus griseus* (Stone goby) spend their adult stages in freshwater: they spawn in freshwater but their larvae migrate to saltwater for growth and maturation [81]. Continuity of their life cycle depends on unimpeded river passage, presence of natural riparian vegetation cover, unmodified flow regimes, low silt load, high-quality estuarine habitats, and absence of introduced species [81].

*Schismatogobius deraniyagalai* (Redneck Goby, Figure 5), a goby restricted to open, clear, shallow waters with a sandy substrate (HS, personal observations) is vulnerable to impaired water quality and sand mining along the river. At least 25 species of alien freshwater fish are established in Sri Lanka, exerting pressure on native freshwater biodiversity, and the Kelani River is no exception [82]. This basin, particularly its urban and residential reaches, hosts several alien fish species including *Clarias batrachus* (Walking catfish), *Chitala ornata* (Clown featherback), *Pterygoplichthys* cf. *disjunctivus* (Vermiculated Sailfin Catfish), *Poecilia reticulata* (Guppy), *Oreochormis mossambicus* (Mozambique Tilapia), *O. niloticus* (Nile Tilapia), *Trichopodus pectoralis* (Snakeskin gourami), *T. trichopterus* (Three-spot gourami), and *Helostoma temminkii* (Kissing gourami) [20,83–85]. Moreover, other aquatic exotic invaders—*Trachemys scripta* (Red-eared slider, a turtle) and *Pomacea diffusa* (Apple snail) in particular—have been documented along the lower reaches of the river (HS, personal observations). These alien species compete with native fishes for critical resources such as food, habitats, and nesting grounds (*Oreochromis* species) [22]. Predatory alien fish (*Clarias batrachus* and *Chitala ornata*) prey on small-sized native fish, while *Poecilia reticulata* feeds on native amphibian eggs [84]. These exotic species can change habitat structure and ecological processes, rendering habitats unsuitable for native freshwater biota. For instance, the invasive catfish *Pterygoplichthys* cf. *disjunctivus* burrows nests in the river banks and induces bank erosion and instability; this catfish is also known to alter benthic habitat structure, reduce availability of benthic resources, and modify nutrient dynamics [86]. *Macrognathus pentophthalmos* (Lesser Spiny Eel), a species now extremely rare in Sri Lanka, used to be once common in the lowland flood plains of Kelani River basin such as at Hanwella, where it had been harvested in the fishery [87]. The combination of several factors, such as invasive predatory species and pollutants, is likely responsible for the rarity of

this species at present. Certain aquatic invasive plants (water hyacinth, *Eichhornia crassipes*) increase sediment retention and organic-matter accumulation in water, impairing water clarity and dissolved oxygen levels while increasing evaporation rates [84]. Many urban wetlands (Figure 6) of Kelani River basin, particularly those located in metropolitan Colombo, have been prolifically invaded by alien aquatic plants, rendering dramatic changes in habitat structure and wetland hydroperiod [85].

## 4. Future Directions for Conservation and Management of Kelani River Basin

Kelani River is among the most impacted river basins in Sri Lanka due to unprecedented human activities [9]. Thus, conservation and management of its aquatic resources and biodiversity is imperative and integral to nation's socioeconomic development.

### 4.1. Aquatic Biodiversity Conservation

There are 25 established state-owned protected areas and 15 proposed protected areas, which only protect 9.4% (238.54 km$^2$) of the Kelani River basin [88] (Figure 7). The management objectives of the existing protected areas of the basin should include watershed-scale conservation planning, maintaining habitat structure and ecosystem functions along the entire riparian corridor. Future reserve selection, conservation prioritization, and ecological restoration in the Kelani River basin should focus on (1) species richness and endemism of freshwater fishes, rarity of aquatic and semiaquatic species, and presence of threatened aquatic or semiaquatic fauna and flora [89–91]; (2) wetlands and aquatic habitats susceptible to human-induced disturbances [90–93]; (3) management of erosion-prone landscapes such as high-gradient slopes and historically-logged areas [9]; (4) riparian buffer conservation to ensure spatially-continuous and functionally-connected riverscapes and wetland complexes [94–96]; and (5) scenic, aesthetic, and cultural values of riverscapes [97,98]. Given that the riparian zone is either severely degraded or nonexistent in urban areas, reforestation of the riparian buffer zones is necessary, at least in the urban segments of the river.

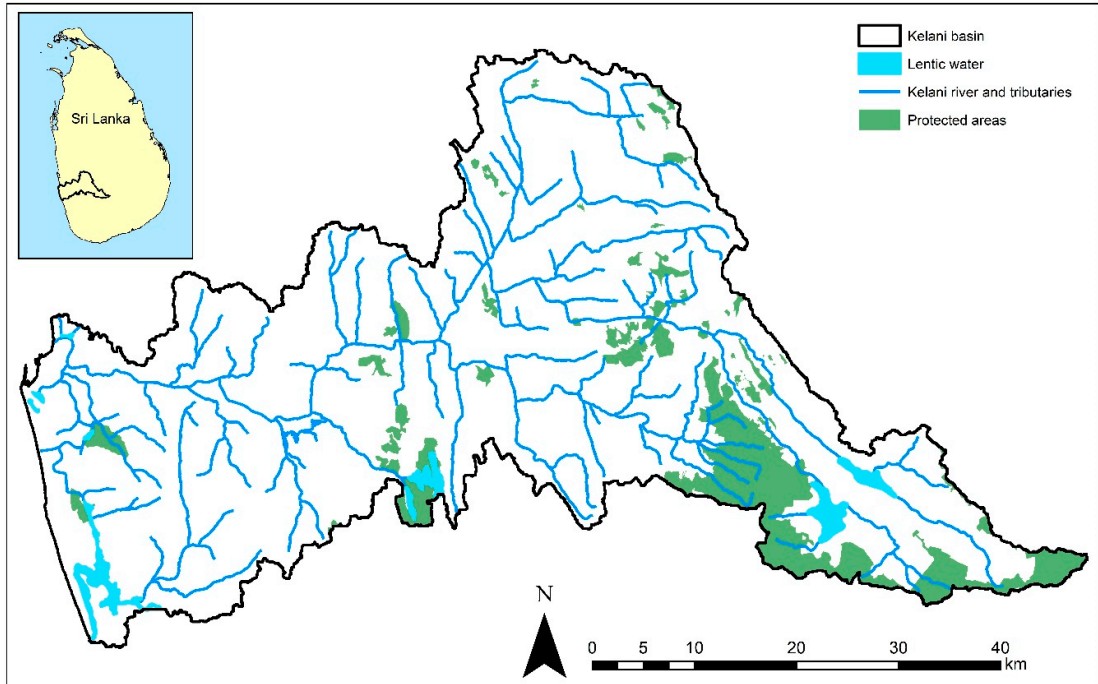

**Figure 7.** Kelani River Basin with the main channel and other tributaries with distribution of large lentic water bodies and protected areas. Data sources: World Database on Protected Areas [88], International Water Management Institute [99].

Due to resource limitations and conflicts in land management objectives, declaring conservation lands is an insurmountable challenge in Sri Lanka. This predicament is much severe in diversity rich, rapidly developing riverscapes like Kelani River. Therefore, conventional "fortress conservation" needs to be complemented with novel approaches such as conservation easements for upper watersheds and floodplains, riparian conservation reserves, trees-outside-forests, demarcation of urban parks and town conservation commissions, and delineation of freshwater protected areas [12,100,101]. Organic farming practices, conservation tillage practices, soil conservation measures, and agroforestry may enhance stream water quality and supply woody debris and allochthonous energy to the stream channel.

Restoration is a crucial element in Kelani River basin conservation [73]. Foremost, specific parts of Kelani River in need of restoration should be prioritized. Efforts should focus on establishing meandering and braiding flow pathways, restoring aquatic and wetland habitats of the floodplains, bank stabilization and erosion control, reforestation of riparian zones, introduction of large woody debris into the river channel to restitute channel complexity, and reconnecting the river channel with its floodplains [5,10,58,96,102–104]. We also cautioned against reclamation of wetlands into lakes and ponds by dredging or channelizing lotic systems which has been the norm for urban and amenity-based development throughout Kelani River basin. Moreover, introduction of alien flora should be avoided for landscaping or horticulture near Kelani River since watercourses act as dispersal pathways for invasive species [84,85].

Distribution and taxonomy of Sri Lankan freshwater fish have been well established [105–109], thus makes a key biodiversity element of Kelani River. Several freshwater fish of Kelani River have narrow distribution ranges (e.g., *Pethia bandula* and *Systomus asoka*) and a few are migratory species (e.g., *Sicyopus jonklaasi* and *Sicyopterus griseus*) that use different stream orders for a variety of life-history functions. The persistence of Kelani River's native ichthyofauna depends on the optimal habitat quality of the entire basin, thus serves as an "umbrellas species assembly". Species such as *Laubuka varuna* (Varuna Laubuka), *Pethia nigrofasciata*, and *Systomus pleurotaenia* are sensitive to water quality degradation and habitat perturbation and should be considered "indicator species" [4,12,103]. Given their habitat specialization, indicator fish are also symbolic of unimpaired stream habitats, thus serves as "flagship species" [5]. Further, fish (*Sicyopus jonklaasi*, *Pethia nigrofasciata*, *Pethia reval*, and *Aplocheilus dayi*) that possesses unique body forms and colorations make ideal charismatic species.

Conservation of Kelani River biodiversity should focus on functional integrity and ecosystem services, including flood mitigation, regional climate moderation, resilience to climate change, providing water for domestic and commercial needs, hydropower generation, leisure, and inland capture fishery [1,10,104]. Preserving functional attributes of Kelani River require troubleshooting a number of legislative issues, yet, yields lasting solutions since ecosystem services are tangible commodities marketable to policy makers, general public, and exploitative entrepreneurs [10]. We strongly encourage evaluation of Kelani River as an economic asset by assigning a "dollar value" to its ecosystem services [110,111].

## 4.2. Future Research

Scientific knowledge on ecological attributes of its biota is imperative for effective conservation of Kelani River [89]. For instance, identifying obligatory migrants, narrow-ranging species, age-structured niche partitioning, and habitat preferences help understand species vulnerabilities to habitat modifications [5]. Long-term assessments on population status, modifications in species distribution ranges, metacommunity dynamics, and ecosystem functions of river biota should be studied in the context of changes in basin-wide land uses, water quality, and in-stream habitat structure so that drivers of ecological change can be clearly identified. Data on population status and range decline of rare and endemic species are a gap in current knowledge in aquatic biodiversity of Sri Lanka; filling-in such gaps is critical for conservation assessments and subsequent species-specific or habitat management plans. Following long-term monitoring standards, we propose setting up permanent survey stations throughout the basin to study changes to both aquatic biodiversity and water quality.

To quantify habitat quality of Kelani River, we propose an index of biotic integrity based on species richness, functional attributes, and habitat specialization of freshwater fish [112–114]. This index will help conservation authorities detect environmental damage prematurely so that timely remedial actions can be taken. Effectiveness of restoration actions and habitat management efforts can also be evaluated via this index.

Reproductive biology of Kelani River's native fish, particularly that of narrow-range endemics, is not well understood. Consequently, a few attempts on captive breeding of have ended in failure [79]. Research on breeding biology of Kelani River's narrow endemics can inform ex-situ conservation, which is paramount for their persistence as their natural habitats continue to deteriorate.

There are several exploitative lobbies that have operated in Kelani River for several decades such as capture fisheries, aquaculture, and collections of ornamental fish (personal observations); yet, the ecological impacts of these exploitations are unknown. Inland capture fisheries have been a lifelong tradition in Sri Lanka, but, these practices mostly concentrate on lentic systems and large rivers of Sri Lanka's dry zone. Most indigenous fish of Kelani River are too small to have any food value, thus, are less likely to be harvested. However, certain broadcast fish capture techniques—use of underwater explosives, electroshocking (dropping high-voltage live wires into water), use of plant-extracted paralytic agents, and casting small-meshed gillnets—can increase the bycatch [42]. The extent to which these activities are practiced in Kelani River and their impacts needs further investigation.

Collecting freshwater fish from the wild for international aquarium pet trade has been a serious issue in southeast Asia [115]. However, the impacts of harvesting indigenous ornamental fish in Sri Lanka needs to be studied. Among exported ornamental freshwater fish of Sri Lanka, 98% comes from wild stocks [116]. A number of heavily exploited ornamental fish of Sri Lanka—particularly *Garra ceylonensis*, the most exploited species—is found in Kelani River [116]. Population assessments of ornamental freshwater fish in Kelani River basin should be implemented as the ornamental fish trade gains popularity as a lucrative business [115]. Feasibility of hatchery rearing and captive breeding of ornamental fish should be explored and shared with fish exporters [115]. These research studies should also inform policy decisions and reformations at least in three fronts: (1) identify most exploited populations to impose harvest bans, (2) identify range-restricted species and species with small or declining populations in the Kelan River basin where harvesting can be unsustainable, and (3) developing a scientifically formulated platform to petition rare and threatened fish species of Kelani River basin in CITES appendices [116].

Possible impacts of climate change across the basin is also understudied. Increase in extreme precipitation and temperatures are predicted for the mid basin coupled with dramatic changes in the overall discharge [38]. However, such analyses are limited to smaller catchments, thus warrants basin-wide modelling. The biological responses of such hydrological and climatic shifts are unknown.

We urge the need to publish research in peer reviewed indexed journals. Over 20% of scientific research pertinent to biodiversity, conservation, and water quality of Kelani River is found in either technical reports or in conference proceedings. Much data on water quality assessments, policy concerns, and management actions are overwhelmingly found in such grey literature. These studies followed rigorous and standard research protocols, yet, were not published in journals. To garner international interests and support for conservation and management of Kelani River basin, we underscore the need to elevate research into publication standards. In the developing tropical realm, lack of a scientific foundation has greatly impeded formation of sound policies.

## 5. Conclusions

Kelani River basin is undergoing rapid industrialization as well as commercialization, thus, confronts the most pressing water-quality and other environmental issues in Sri Lanka, including loss of biodiversity. Serious ecological collapse and public health complications are inevitable unless science-based resource management strategies and aquatic biodiversity conservation actions are taken. Kelani River conservation programs must be comprehensive and should be reinforced through

integration of stakeholder partnerships and participatory management. These multilateral partnerships should cultivate multipurpose sustainable management of aquatic resources where conservation efforts and wise use of resources complement each other. The environmental status and rich biodiversity associated with Kelani River basin provide a unique opportunity for development of sustainable water resource conservation and management in Sri Lanka. Conservation potential and biodiversity richness of Kelani River basin are comparable to many other tropical riverscapes, particularly in south and southeastern Asia. Therefore, conservation challenges and policy issues we discussed pertain to other tropical rivers and riverine ecosystems; as such, our recommendations and guidelines will assist conservation planning in other biodiversity rich tropical river basins.

**Author Contributions:** Each author made substantial contributions to the conception of the work, including drafting the manuscript and in subsequent revisions; and all authors have approved the submitted version; and agree to be personally accountable for their own contributions. Conceptualization, R.K., S.K., and T.S.; Validation, H.S. and S.K.; Writing—Original Draft Preparation, T.S.; Writing—Review and Editing, T.S., S.K., H.S., and R.K.; Visualization, T.S., H.S. and SK. All authors have read and agreed to the published version of the manuscript.

**Funding:** This research received no external funding and the APC was funded by Bridgewater State University.

**Acknowledgments:** The authors would like to thank Rohan Pethiyagoda for commenting on the manuscript and Shantha Jayaweera, Chamara Amarasinghe, Indika Peabotuwage, Shanaka Lakmina, Supun Lahiru Prakash, Mahesh De Silva, Tharaka Kusuminda, Sameera Akmeemana, Krishan Wewalwala, Madhava Botejue and Dinesh Gabadage for information sharing and providing photographs.

**Conflicts of Interest:** The authors declare that they have no conflict of interest.

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
