# Peer review of "Challenges in Biodiversity Conservation in a Highly Modified Tropical River Basin in Sri Lanka"

_water, doi:10.3390/w12010026_

Round 1

Reviewer 1 Report

The ms presents an assessment of the Kelani river catchment area and proposes legislative changes to improve conservation and management.  It references a similar study, Env. Auth & IUCN 2016 [9] without identifying how this study adds value.  The ms wavers between being a scientific review of the ecological status of the river system, and a policy document for managing the basin.  It should focus on the former.

An environmental literature review of the river catchment system is valuable and I recommend this ms be considered again after major review.  The authors should rewrite with a focus on (1) what is known to be true (scientifically verified), and (2) knowledge gaps (not scientifically verified, even if it has been widely commented on in policy literature).  They should focus primarily on journal publications.  They can describe current and past policy / law and to identify weaknesses (e.g.; evidence of lack of enforcement) but should not propose specific policy changes.  A history of basin management is also useful.

I went through to line 164 in detail (below) but then stopped, since it became clear to me that a major rewrite is necessary to tighten the scientific rigor of the ms

Writing style:

Introduction is well written, but all other sections including the abstract need to be tightened.

L74-77 needs rewriting. Change ‘compressive’ to ‘comprehensive’.  Perhaps:

In this study we review indigenous freshwater fish diversity and prevailing threats, and provide recommendations for conservation and management of the Kelani river ecosystem, based on published literature and author observations.

Fig 1: I’d change the terms ‘head waters, mid reaches, lower reaches’.  Perhaps ‘upper, mid, lower basin’.  Consider overlaying a contour map onto this fig

Fig 1: Title needs to be strengthened.  Be careful of over-capitalization.

Fig 2: Remove top three items on legend (above land cover types)

L84: change to ‘Threats’

L86: change heads to head.

L88: ‘bombarded’, ‘tremendous’ are subjective words and should therefore be removed.

L88: growing fast – upwards, outwards, or both?

L85-89 – I think this section should be under 2.1

L90 – This section needs significant restructuring.  A suggested structure is below.

2.1 Land use change

2.1.1 Urbanisation

              (Affected mostly the lower basin region)

2.1.2 Annual crops

              (here I think you are also talking mostly about the lower basin.  In L99-100 you mention vegetation structure

2.1.3 Perennial crops

              (tea)

2.2 Pollution

2.2.1 Point source

2.2.2 Non-point source

Additionally, section 2.1 appears academically weak to me.  There must be a great deal more information about land use change in the different regions of the river.  The reference to vegetation structure (line 99-100) appears too anecdotal.  What is the evidence for vegetation change?  Large trees can be seen on Google Earth, which has free data from last year and from 2001 / 2004.  I had a brief look and found riparian areas where trees had increased, and others where they had decreased, over this period.  Hence it is possible, free, and not very difficult, to quantify tree change in different areas of the basin over the last 15-20 years.  Likewise, forest clearing for paddy could also be estimated.  The authors should do this, or include other references of land use change that have done something similar.

L139.  This paragraph also needs tightening.  The authors have cited a conference presentation, a river management plan, and a USDP project to support the notion that 6000+ industries are directly polluting the river.  This may be (is probably) true, but it’s not scientific.  Have water quality studies been conducted? Has research been conducted on industrial wastewater?  Authors need to either include more scientific journal references on these topics, OR specify that the relevant research hasn’t been done but previous development reports have suggested / concluded that …

The authors have specified some examples of contaminants, but should recognize that pretty much everything is being poured into this river system.  It is unwise to try to list the range of contaminants since a complete assessment would go on forever.  Instead, it would be more useful to describe the research that has been conducted and place it better in a temporal framework (e.g.; a 1985 study identified a diverse range of biologically hazardous contaminants, including heavy metals, oils, hydrocarbons and polychlorinated biphenyls [51])

Some information on water flows would also be useful.  How long does it take water to reach the ocean? (affects how quickly pollutants are removed from the system)

L193.  I suggest adding a table that shows

Law Issue being addressed Date last reviewed Enforcement (who is responsible for monitoring)

Authors need to replace opinion statements (e.g.; effectiveness of those policies is dubious) with verifiable statements (e.g.; water quality has remained at …)

L613 – ref 24 – journal?

L645 – information, not information.  The link appears to be broken

Author Response

Reviewer one General comments

Reviewer comments (RC): The ms presents an assessment of the Kelani river catchment area and proposes legislative changes to improve conservation and management.  It references a similar study, Env. Auth & IUCN 2016 [9] without identifying how this study adds value.  The ms wavers between being a scientific review of the ecological status of the river system, and a policy document for managing the basin.  It should focus on the former.

Author Response (AR): we agree. We have now focused our the ms on reviewing ecological status of the Kelani river and shifted away from the policy focus. As a result, we have cut down the text length significantly. Nonetheless, we do not agree with complete removal of all policy elements. Specific comments did not encourage elimination either. We refrained from providing a very specific policy framework—rather limited ourselves to point out weaknesses in current policies and provide some basic guidance on rectifying them. The current revisions will also make the ms broader in appeal too.

RC: An environmental literature review of the river catchment system is valuable and I recommend this ms be considered again after major review.  The authors should rewrite with a focus on (1) what is known to be true (scientifically verified), and (2) knowledge gaps (not scientifically verified, even if it has been widely commented on in policy literature).  They should focus primarily on journal publications.  They can describe current and past policy / law and to identify weaknesses (e.g.; evidence of lack of enforcement) but should not propose specific policy changes.  A history of basin management is also useful. I went through to line 164 in detail (below) but then stopped, since it became clear to me that a major rewrite is necessary to tighten the scientific rigor of the ms

 AR: Again, we agree. We focus the current ms on reviewing published peer-reviewed literature, conference proceedings, and included some technical reports as well. We also highlighted the knowledge gaps, mostly under “Future directions for conservation and management of Kelani River basin”. Throughout the ms, particularly in sections 1, 2, and 3 we relied primarily in journal articles. However, there is much information locked in grey literature that have followed rigorous sampling protocols but for some reason did not end-up in a journal. While we certainly have eliminated much of grey literature, we retained some of them. As suggested, in section 2.4, 3.1, and 4.3, we highlighted policy weaknesses—but, refrained from creating specific policies.

RC: Introduction is well written, but all other sections including the abstract need to be tightened.

AR: With our revisions, the abstract + text + references of the ms is now under 9000 words. Prior to revisions, this was over 10,600.  We have also read the entire ms several times to reduce the length.

Reviewer one Specific comments

RC: L74-77 needs rewriting. Change ‘compressive’ to ‘comprehensive’.  Perhaps:

In this study we review indigenous freshwater fish diversity and prevailing threats, and provide recommendations for conservation and management of the Kelani river ecosystem, based on published literature and author observations.

AR: suggested changes are now made in line 72-73

RC: Fig 1: I’d change the terms ‘head waters, mid reaches, lower reaches’.  Perhaps ‘upper, mid, lower basin’.  Consider overlaying a contour map onto this fig. Fig 1: Title needs to be strengthened.  Be careful of over-capitalization.

AR: suggested changes in the terminology are now included. Contours were also added to the same map. Capitalization is a requirement of the journal.  

RC: Fig 2: Remove top three items on legend (above land cover types)

AR: we disagree: the lentic and the lotic water bodies should be represented in the context of the basin wide land-use land cover types. Without water bodies, the just land-use distribution across the basin does not provide much information on HOW the river is affected by land-use types.

RC: L84: change to ‘Threats’

AR: Agreed and changed.

RC: L86: change heads to head.

AR: changed

RC: L88: ‘bombarded’, ‘tremendous’ are subjective words and should therefore be removed.

AR: we agree, throughout the ms, we removed similar subjective adjectives, and even restructured the entire sentence to make it more objective.

RC: L88: growing fast – upwards, outwards, or both?

AR: mostly, this is outward growth, and this is now clarified in the ms line 103. 

RC: L85-89 – I think this section should be under 2.1

AR: This part is not added to section 2.1 with significant changes in the text.

RC: L90 – This section needs significant restructuring.  A suggested structure is below.

 2.1 Land use change

2.1.1 Urbanisation

              (Affected mostly the lower basin region)

2.1.2 Annual crops

              (here I think you are also talking mostly about the lower basin.  In L99-100 you mention vegetation structure

2.1.3 Perennial crops

              (tea)

2.2 Pollution

2.2.1 Point source

2.2.2 Non-point source

AR: This is an excellent suggestion. We agree. We included the suggested restructuring to sections 2.1 and 2.3. we also used the same suggestions to revise sections 3 and 4 as well.

RC: Additionally, section 2.1 appears academically weak to me.  There must be a great deal more information about land use change in the different regions of the river.  The reference to vegetation structure (line 99-100) appears too anecdotal.  What is the evidence for vegetation change?  Large trees can be seen on Google Earth, which has free data from last year and from 2001 / 2004.  I had a brief look and found riparian areas where trees had increased, and others where they had decreased, over this period.  Hence it is possible, free, and not very difficult, to quantify tree change in different areas of the basin over the last 15-20 years.  Likewise, forest clearing for paddy could also be estimated.  The authors should do this, or include other references of land use change that have done something similar.

AR: This is a useful comment. However, information available on precise land-use changes are meager. Through GEE and other sources, we were able to quantify the current land-use conditions. But, GEE or other free remote-sensing data bases do not have adequate coverage for this basin going beyond just a couple of years. The google earth imageries in 10 yrs or more into the history, if present, are incomplete and of much lower resolution. After some exhaustive efforts, we were able to gather some information to quantify land-use change, these are now included into different sections of 2.1. With much efforts, we were able to provide some reliable quantification for land-use change and omitted any anecdotal references.  

RC: L139.  This paragraph also needs tightening.  The authors have cited a conference presentation, a river management plan, and a USDP project to support the notion that 6000+ industries are directly polluting the river.  This may be (is probably) true, but it’s not scientific.  Have water quality studies been conducted? Has research been conducted on industrial wastewater?  Authors need to either include more scientific journal references on these topics, OR specify that the relevant research hasn’t been done but previous development reports have suggested / concluded that …

AR:  We agree and understand that we should primarily adhere to citing and referencing journal manuscripts. Unfortunately, much information is stuck in non-journal literature. As most of this info comes from governmental technical reports, the information conveyed are reliable. However, we cited several more recent journal articles to validate these points under all sections for 2.2. Following the reviewer suggestions, we underscored the need of research and published primary literature in section 4.4.    

RC: The authors have specified some examples of contaminants, but should recognize that pretty much everything is being poured into this river system.  It is unwise to try to list the range of contaminants since a complete assessment would go on forever.  Instead, it would be more useful to describe the research that has been conducted and place it better in a temporal framework (e.g.; a 1985 study identified a diverse range of biologically hazardous contaminants, including heavy metals, oils, hydrocarbons and polychlorinated biphenyls [51])

AR: We somewhat disagree. It is important to understand what specific contaminants are present in water as different contaminants have different sources and need different treatments for rectification. The list of contaminants are mostly listed in a table, not in a text so that the text is not burdening to the reader. However, we do agree with highlighting the temporal framework, particularly in issues that have been documented historically and persisting. We corrected the sentence structure to reflect the time base in sections 2.2 and 2.3.        

RC: Some information on water flows would also be useful.  How long does it take water to reach the ocean? (affects how quickly pollutants are removed from the system)

AR: data on discharge is now added to section 2.2 in the context of pollution.

RC: L193.  I suggest adding a table that shows Law Issue being addressed Date last reviewed Enforcement (who is responsible for monitoring) Authors need to replace opinion statements (e.g.; effectiveness of those policies is dubious) with verifiable statements (e.g.; water quality has remained at …)

AR: This is an excellent suggestion. We added the table (now table 2) with the key information on policies. We also omitted subjective statements.   

RC: L613 – ref 24 – journal?

AR:  this reference is a journal, and corrected in the bibliography.

RC: L645 – information, not information.  The link appears to be broken

AR: link repaired, and spelling corrected.

Reviewer 2 Report

・General comment

This paper deals with the concept of biodiversity conservation and management in the modified watershed belonged in the south Asia. The current situation and problems of the watershed was well organized and this paper proposed the important information. Although I have no doubt about the quality of the presented work, I recommend to emphasize the generality as can be applied to other watersheds with similar challenges. I have found a few issues that, once addressed, will improve the manuscript.

・Specific comment

The importance of this watershed from the view point of biodiversity should be described.

Specific data on change of fish fauna or decreasing rare species should be added.

It is easy for readers to understand by listing the anthropogenic impacts that affect the biodiversity and showing recent changes with specific date.

Collaboration between diverse actors is important for biodiversity conservation, however, if there are any barriers to build consensus in this watershed, please discuss them.

Resolution of figures should be improved.

Author Response

Reviewer two General comments

RC: This paper deals with the concept of biodiversity conservation and management in the modified watershed belonged in the south Asia. The current situation and problems of the watershed was well organized and this paper proposed the important information. Although I have no doubt about the quality of the presented work, I recommend to emphasize the generality as can be applied to other watersheds with similar challenges. I have found a few issues that, once addressed, will improve the manuscript.

AR:  Thank you for this encouraging remark. We have underscored the general applications of our findings in the manuscript.

Reviewer two specific comments

RC: The importance of this watershed from the view point of biodiversity should be described.

AR: under section 3, we added a new paragraph to hammer this point. Section 3.1 and 4.3 also supports this point as well.

RC: Specific data on change of fish fauna or decreasing rare species should be added.

AR: while we 100% agree that this is a very useful technical comment. Unfortunately, no quantitatively or qualitatively robust data is available for population decline or range shifts. Following this recommendation, we underscored the need to follow research on this theme in section 4.4   

RC: It is easy for readers to understand by listing the anthropogenic impacts that affect the biodiversity and showing recent changes with specific date.

AR: Again, while we admit this is fundamentally correct. But, we do not have any such robust documentation to follow such changes in biodiversity. However, whenever we can track some temporal pattern, we added that—for example, in sections on pollution, land-use change, and hydrologic modifications. However, we do not have corresponding data to ascertain quantified decline. In addition, we hammered the importance of such rigorous documentation in section 4.4     

RC: Collaboration between diverse actors is important for biodiversity conservation, however, if there are any barriers to build consensus in this watershed, please discuss them.

AR: This is excellent point. We have identified such conflicts in section 2.4

RC: Resolution of figures should be improved

AR: High-res images are now supplied separately to the publisher.  

Round 2

Reviewer 1 Report

accept after language proofing. I have submitted comments already

Author Response

Thanks for your comments.

Reviewer 2 Report

The manuscript has been revised well. I think this manuscript is acceptable.

Author Response

Thanks for your comments.